# Effect of High Hydrostatic Pressure Processing on the Chemical Characteristics of Different Lamb Cuts

**DOI:** 10.3390/foods9101444

**Published:** 2020-10-12

**Authors:** Kevin Kantono, Nazimah Hamid, Indrawati Oey, Yan Chao Wu, Qianli Ma, Mustafa Farouk, Diksha Chadha

**Affiliations:** 1Department of Food Science and Microbiology, Auckland University of Technology, Auckland 1010, New Zealand; kkantono@aut.ac.nz (K.K.); ycw@aut.ac.nz (Y.C.W.); mql201228@163.com (Q.M.); diksha.chadha@aut.ac.nz (D.C.); 2Department of Food Science, University of Otago, Dunedin 9016, New Zealand; indrawati.oey@otago.ac.nz; 3Riddet Institute, Massey University, Palmerston North 4474, New Zealand; mustafa.farouk@agresearch.co.nz; 4Institute of Food Science and Technology, Chinese Academy of Agricultural Sciences, Key Laboratory of Agro-Products Processing, Ministry of Agriculture and Rural Affairs, Beijing 100193, China; 5AgResearch MIRINZ, Ruakura Research Centre, Private Bag 3123, Hamilton 3240, New Zealand; mustafa.farouk@agresearch.co.nz

**Keywords:** high-pressure processing, lamb cuts, fatty acids, amino acids, lipid oxidation

## Abstract

The non-thermal high-pressure processing (HPP) technique has been used to increase the shelf life of food without compromising their nutritional and sensory qualities. This study aims to explore the potential application of HPP on New Zealand lamb meat. In this study, the effect of HPP, at different pressure treatments (200–600 MPa) on eight different lamb meat cuts in terms of lipid oxidation, fatty acid and free amino acid content were investigated. In general treatments between 400 and 600 MPa resulted in higher oxidation values in eye of loin, flat, heel, and tenderloin cuts. Saturated and monounsaturated fatty acid content were significantly lower with HPP treatment of almost all cuts (except rump and heel cuts) at all pressures. Polyunsaturated fatty acid content was significantly lower in HPP-treated inside, knuckle, and tenderloin cuts at 600 MPa compared to control. Nine essential free amino acids (valine, leucine, isoleucine, methionine, phenylalanine, lysine, histidine, tyrosine and tryptophan), and eight non-essential free amino acids (alanine, glycine, threonine, serine, proline, aspartic acid, glutamic acids and ornithine) were identified in the lamb cuts. HPP increased the total free amino acid composition significantly compared to control at all pressures for almost all cuts except the inside and eye of loin cuts. This study suggests that higher pressure treatments (i.e., 400 and 600 MPa) resulted in higher TBARS oxidation levels. Additionally, significant decreases in saturated and monounsaturated fatty acids and increase free amino acid content were observed in the majority of HPP-treated samples compared to control.

## 1. Introduction

Lipid oxidation can adversely affect the quality of meat and meat products during processing and may confer negative effects on colour [1], flavour [2], and nutrition [3]. Hence, with the use of high pressure to improve the functional properties and quality of lipids and meat products [4,5], monitoring lipid oxidation in high-pressure-treated meat is necessary for successful implementation of HPP technology in the meat industry. Increased oxidation of pressure-treated pork was significant at high pressures that exceeded 300MPa [6]. Angsupanich and Ledward [7] also reported an increase in lipid oxidation at 400 MPa and higher pressures at ambient temperature in cod muscle. It was further found that the pressure required to initiate lipid changes in turkey [8] and beef muscle was lower (200 MPa) [9] than that required for pork and chicken. Beltran et al. [8] reported that pressure treatment of minced chicken breast at pressures up to 500 MPa had no significant effect on lipid oxidation. Chicken breast meat was found to have more oxidative stability than turkey muscle. Pressure was also found to induce lipid oxidation during subsequent storage of meat [8]. Wang et al. [10] evaluated the effect of high pressure treatment on lipid oxidation and composition of fatty acids in yak body fat at 4 and 15 °C for up to 20 days of storage. At the end of storage, 400 and 600 MPa treatments increased the level of thiobarbituric acid reactive substances (TBARS) by 335% and 400%, respectively, indicating increased lipid oxidation [10]. The mechanism that induces lipid oxidation during HPP treatment is little understood. It has been reported that the release of *haem* molecules through membrane disruption can trigger lipid oxidation [11].

The fatty acid composition of lipids can influence meat quality [12]. Fatty acid composition contributes to quality traits of meat, such as nutritional value, as well as flavour, and textural properties. It varies widely depending on species, degree of trimming, nature of processing or cooking, and preservation techniques employed [13]. Schmid [3] indicated that most common meats (lamb, beef, pork) have a similar proportion of saturated (45% to 50%) and monounsaturated fatty acids (38% to 43%) and provide a small quantity of polyunsaturated fatty acids (12% to 20%). Recent studies have investigated the effects of HPP on fatty acid composition. Most researchers [14,15,16] found that HPP treatment in meat had no significant effects on the fatty acid composition of the total lipids. However, that does not necessarily mean it did not have any significant effect on the phospholipids, triglycerides and free fatty acids. In most studies, HPP-treated samples generally oxidised more rapidly, with little changes in fatty acid composition. However, the critical pressures that influence oxidation can vary with food type. Yagiz et al. [16] reported no significant differences between control and HPP-treated Atlantic salmon dark muscle in terms of total saturated fatty acids (SFAs), monounsaturated fatty acids (MUFAs), as well as n-3 and n-6 polyunsaturated fatty acids (PUFAs). Kang et al. [17] similarly reported no significant differences in the fatty acid content of control and HPP-treated Korean black goat meat samples. Only Wang et al. [10] reported significantly low PUFAs in pressure-treated yak body fat with 600 MPa treatment. McArdle et al. [18], on the other hand, reported significantly higher PUFA/SFA ratios of pressurised samples compared to control samples, with the exception of milder treatments (20 °C at 200 and 400 MPa). Ono et al. [19] only reported an increase in the PUFA/SFA ratio of cooked beef samples compared to raw meat.

Muscle proteins are susceptible to oxidative reactions that may result in loss of essential amino acids and a decrease in protein digestibility. Although the mechanisms and reaction pathways for lipid and protein oxidation are different, they are both influenced by similar prooxidant and antioxidant factors [20]. High pressures between 100 and 300 MPa for 10 min at 25 °C increased the overall autolytic activity of raw beef meat, resulting in a higher concentration of free amino acids [21]. Suzuki et al. [2] reported that serine, glutamic acid, glutamine, glycine and alanine content in lean beef meat gradually increased with increasing pressure, up to 200 MPa. However, no significant differences in all 10 amino acids investigated were observed, suggesting that high-pressure treatment did not influence amino acid composition. Campus et al. [22] investigated the effect of HPP (300, 350, and 400MPa) on sliced- and vacuum-packaged commercial dry-cured pork loin. Only untreated samples showed an increase in free amino acid content during vacuum storage. In fact, application of high pressures (300 to 400 MPa for 10 min at 20 °C) was reported to stabilize free amino acid content during storage due to a reduction in amino peptidase activity.

Although researchers have assessed the impact of HPP on meat quality, limited research has been carried out to investigate the effect of HPP on different lamb meat cuts. The aim of this study is to investigate the effects of high hydrostatic pressure processing (HPP) treatments (200–600 MPa) of eight different lamb cuts on lipid oxidation, fatty acid and free amino acid content. This would help provide an understanding of the chemical changes that may influence the quality of different lamb cuts with HPP treatments.

## 2. Materials and Methods

### 2.1. Preparation of Lamb Samples

Eight meat cuts, inside (M. semimembranosus), heel (M. gastrocnemius), knuckle (M. quadriceps femoris), rump (M. gluteus medius), tenderloin (M. psaos major), eye of loin (M. longissimus), bolar (M. infraspinatus) and flat (M. bicep femoris) were obtained from six lambs (cold carcasses with a weight of 140.5 to 150.5 kg) at 48 h postmortem. The animals were obtained from AgResearch (Hamilton), and carcass cutting, packaging as well as freezing were carried out there. Each muscle was divided into five blocks and vacuum-packed in polyethylene plastic bags labelled: Control, HPP-200, HPP-300, HPP-400 and HPP-600, and immediately frozen at −18 °C in a digitally temperature-controlled thermostat freezer with a static flow cold air temperature of −20 °C for HPP processing. Samples were thawed overnight at 4 °C prior to HPP treatment. In summary, eight different muscles (n_muscle_ = 8) were divided into five separate parts (n_parts_ = 5) from a sample population of six lambs (n_lambs_ = 6); therefore, a total of 240 samples were retrieved, and their physicochemical properties were then analysed in triplicate.

The lambs were all rams from a Coopworth base/composite flock; from weaning, they were run in one mob under commercial conditions on pasture. Lambs were approximately 32 weeks of age at slaughter. The lambs were slaughtered under commercial conditions at a New Zealand meat processing plant (average live weight of 44 kg). Pilot testing was also carried out to investigate the variance between samples. Our pilot results suggest that the variances between animals were small and did not reach statistical significance. HPP processing parameters were selected based on the ranges that had been used in previously published studies.

### 2.2. HPP Processing

Each of the eight different frozen lamb cuts that were vacuum-packed were thawed overnight prior to HPP treatment. Pressurization of lamb was conducted using an industrial scale HPP equipment (HPP 055, Multivac, Multivac Sepp Haggenmüller GmbH & Co., Wolferschwenden, Germany). Water was used as the pressure-transmitting medium, with the initial temperature around 7–8 °C. The temperature reached after pressure build-up was less than 25 °C. The rate of pressure build-up was conducted at 125 MPa/min. Packaged lamb samples were loaded in a cylindrical loading container and HPP-treated at 200, 300, 400, 500 and 600 MPa. Pressure was held for one minute once the targeted pressure was achieved. After depressurisation, all samples were transported and stored at −20 °C for further analysis.

### 2.3. Lipid Oxidation

The 2-thiobarbituric acid reactive substances (TBARS) method to assess lipid oxidation was carried out according to Nam and Ahn [23]. The method was modified by measuring the amount of malondialdehyde (MDA) present in the sample and carried out as described by Faridnia et al. [24]. Minced meat samples (3.0 g) were homogenised using a homogenizer mixer (Janke Kunkel IKA Labortechnik Ultra Turrax T25) in 9.0 mL deionised distilled water at 14,000 rpm for 30 s. Lamb homogenate (1 mL) was obtained and transferred to a disposable test tube. This was followed by addition of 50 µL of butylated hydroxytoluene (BHT; 7.2% *w*/*v* in ethanol) and 2 mL thiobarbituric acid (TBA)/trichloroacetic acid (TCA) solution (20 mM TBA and 15% (*w*/*v*) TCA). The mixture was vortexed and then incubated in a 90 °C water bath for 30 min until a pink colour was observed. Samples were then cooled down in a water and ice bath for 10 min and centrifuged at 3500 rpm for 15 min at 5 °C. The absorbance of the resulting upper layer was measured at 531 nm using a spectrophotometer (Ultraspec 7000 Pro spectrophotometer, Biochrom Ltd., Cambridge, England, UK) against a blank prepared with 1 mL deionised water and 2 mL TBA/TCA solution. The results obtained was expressed as 2-thiobarbituric acid reactive substances (TBARS) in mg malondialdehyde (MDA) per kg of meat using a tetraethoxypropane (TEP) standard calibration curve.

### 2.4. Fatty Acid Methyl Ester (FAME) Analysis

Quantification of total fatty acids was carried out according to Juárez et al. [25] by acid hydrolysis of lipids in lyophilized samples to release free fatty acids. This is followed by in situ esterification to fatty acid methyl esters (FAMEs), and their extraction into toluene for analysis by gas chromatography (GC). Samples were lyophilized for 48 h until completely dried. Then, approximately 20-mg samples were weighed into 10 mL test tubes, and the weight was recorded. A 10 µL aliquot of 2 g/L tridecanoic acid in toluene was added as internal standard followed by further addition of 490 µL of toluene and 750 µL of freshly prepared 5% methanolic HCl. The mixture was mixed using a vortex, and the headspace of each tube was filled with nitrogen. The tubes were then sealed and incubated in a water bath at 70 °C for 2 h. After tubes were cooled down to room temperature, 1 mL of 6% aqueous K_2_CO_3_ and 500 µL toluene were added. The mixture was vortex-mixed and then centrifuged at 1500 x g rpm for 5 min. The organic phase was then removed using a glass Pasteur pipette for analysis of FAME content.

For fatty acid analysis, a Shimadzu GC-17A gas chromatograph equipped with an FID and a FAMEWAX column (30 m × 0.32 mm × 0.25 µm, RESTEK, Inc., Austin, TX, USA) was used. Nitrogen was used as a carrier gas. The pressure was set to 43 Pa, and the flow rate was 7 mL/min. The oven temperature was held for 5 min at 140 °C, increased to 245 °C at 3.5 °C/min, and held for 3 min at this temperature.

### 2.5. Free Amino Acids

Methanol (1 mL) was used to extract free amino acids (FAA) from freeze-dried meat samples according to Penet et al. [26], with modifications. Meat samples (0.1 g) were weighed into a centrifuge tube, and 1 mL of methanol was added. The mixture was vortexed and then centrifuged at 2000× *g* for 2 min. A commercial free amino acid kit (EZ:faast^TM^, Phenomenex^®^, Torrance, CA, USA) was used to profile amino acids (user manual shown in the Appendix). All steps, including the solid phase extraction (SPE) sample clean-up, elution from SPE sorbent, derivatisation, and analysis, were performed as described in the manual provided. Additionally, 0.2 mM Norvaline in N-propanol solution was used as an internal standard.

Free amino acid derivatives were analysed using the EZ:faast™ GC–MS Free Amino Acid kit (Phenomenex, Torrance, CA, USA). A Shimadzu GC2010 GC, equipped with a ZB-AAA GC column 10 m × 0.25 mm × 0.25 um (Phenomenex, Torrance, CA, USA), was used. The instrument settings used are described in the EZ:faast user manual with modifications. Briefly, the derivatised samples (1 µL) were injected into the GC column. The GC temperature program was set at an initial oven temperature of 120 °C (split ratio of 1:15), raised to 165 °C at a rate of 5 °C per minute, and further increased at a rate of 20 °C per minute to reach 320 °C, where it was held for 1 min. A mixture of 26 amino acids, ranging from 50 to 400 nmol/mL, was used for identification and quantification. The concentration of each amino acid identified in the samples was reported as mg of amino acid in 1 g of lamb sample.

### 2.6. Statistical Analysis

The experimental data in this study was collated using Microsoft Office Excel 2011 and subjected to statistical analysis using XLSAT 2020 (Addinsoft, New York, NY, USA). Shapiro–Wilk’s normality test was applied to check the normality of the data. The data collected in this study were found to be normal; hence, parametric statistical methods were applied. Analysis of variance (ANOVA) was carried out at the 0.05 level of significance to analyse the effect of HPP processing (control, P200, P300, P400 and P600) on lipid oxidation, fatty acid, and amino acid content in 8 different lamb-cut samples (tenderloin (T), rump (R), knuckle (K), inside (I), heel (H), flat (F), eye of loin (E) and bolar (BL) muscles). A two-way analysis of variance was carried out on the fatty acid profiles for each pressure treatment. When ANOVA was significant (*p*-values less than 0.05), means were separated by pairwise comparison using Fisher’s least significant difference test.

Additionally, multiple factor analysis (MFA) was carried out in order to summarise how the different meat cuts and HPP pressure treatments influenced fatty acid and free amino acid content in this study. MFA is a common statistical tool used to analyse multiple data tables that measure sets of variables collected on the same samples simultaneously, which enables the investigation of relationships between datasets for the variables, namely, amino acids, fatty acids, and TBARS. MFA utilises PCA, which was carried out for each variable of the various data tables [27]. The first eigenvalues of each PCA were then used to weight the various tables. After each PCA was carried out, a weighted PCA on the columns of all tables was carried out.

## 3. Results and Discussion

### 3.1. Lipid Oxidation

#### 3.1.1. Lipid Oxidation in Different Cuts of Meat

Lipid oxidation is a very important factor that affects lamb meat quality and acceptance [28]. As lipid content may vary in different animal muscles, the level of lipid oxidation may vary as well. Other factors that can influence the level of lipid oxidation include breed, age, and gender. In this study, the lipid oxidation levels of eight different cuts (flat (F), tenderloin (T), rump (R), knuckle (K), bolar (BL), inside (I), heel (H) and eye of loin (E)) subjected to HHP (0, 200, 300, 400 and 600 MPa) were determined.

In this study, there were significant differences in the overall oxidation level with the eight different cuts. According to Park et al. [29], TBARS values varied with pork meat cuts (belly and loin). Oxidation values of belly cuts were significantly higher than loin. In contrast, Kannan et al. [1] reported no significant differences in overall oxidation level in leg shoulder (cut heel), arm (cut flat), and loin/rib (cut eye of loin) cuts of goat meat.

According to Wood et al. [30], TBARS values above 0.5 mg MDA/kg will produce a rancid flavour that can be detected by consumers. All the values in control cut samples were below 0.5 mg MDA/kg (0.13 to 0.33 mg MDA/kg). Similarly, Rhee et al. [31] reported values between 0.24 to 0.35 mg MDA/kg in beef *longissimus dorsi* (eye of loin), psoas major (tenderloin), *semimembranosus* (inside), and *semitendinosus* (heel) muscles. Figure 1A shows that flat (0.33 mg MDA/kg), tenderloin (0.30 mg MDA/kg) and rump cuts (0.29 mg MDA/kg) had significantly higher TBARS values than other cuts. Heel and eye of loin cuts, on the other hand, had significantly the lowest TBARS value. These findings are supported by Badiani et al. [32], who reported that flat cut had the highest lipid content compared to eye of loin and heel cuts of raw beef muscles. Rhee et al. [31] further reported that the extent of porcine muscle lipids to undergo lipid oxidation may vary substantially among the same retail cuts and different animals even if the postmortem history of the meat is similar. Results of lipid oxidation from this study on different lamb cuts supports this notion. 

#### 3.1.2. Effect of Different HPP Treatments on Lipid Oxidation of Different Lamb Cuts

HPP treatment at higher pressures resulted in some samples having high TBARS values, exceeding 0.5 mg MDA/kg, that can contribute to rancid flavour [30,33,34]. Specifically, the eye of loin cuts (Figure 1B-e) had significantly the highest MDA values when treated at 400 MPa (0.80 mg MDA/kg) and 600 MPa (0.9 mg MDA/kg), followed by flat cut (Figure 1B-f) at 600 MPa (0.8 mg MDA/kg), heel cut (Figure 1B-h) at 600 MPa (0.54 mg MDA/kg), and tenderloin cut (Figure 1B-d) at 400 MPa (0.63 mg MDA/kg) and 600 MPa (0.57 mg MDA/kg). Similarly, McArdle et al. [18] reported a significant increase in TBARS values of lamb brisket cut (M. *pectoralis profundus* muscle) after 30 days’ storage with increased pressures (200, 400 and 600 MPa at 60 °C) during high-pressure treatment. Ma et al. [9] reported that the TBARS value of beef muscle (stored for 7 days at 4 °C and treated at 20 °C) was significantly the highest at 400 MPa (0.67 mg MDA/kg), which then decreased at 600 MPa (0.52 mg MDA/kg) and 800 MPa (0.41 mg MDA/kg). Only knuckle cut (Figure 1B-c) in this study had a TBARS value of above 0.5 mg MDA/kg at 300 MPa (0.52 mg MDA/kg). In general, when higher levels of pressure (400 and 600 MPa) were employed, it resulted in higher oxidation values exceeding 0.5 mg MDA/kg in eye of loin (Figure 1B-e), flat (Figure 1B-f), heel (Figure 1B-h) and tenderloin (Figure 1B-d) cuts, except for knuckle cut (Figure 1B-c) that had the highest oxidation value at 300 MPa. On the other hand, lower pressures of 200 and / or 300 MPa resulted in oxidation values less than 0.5 mg MDA/kg for all cuts. In previous studies [6,9], elevated pressures at room temperature decreased the oxidative stability of red meat. The pressures required to initiate these changes were lower for beef (200 MPa), compared to pork (300 MPa) and chicken (600 MPa), although the postslaughter history of the samples varied [9]. It has been postulated that this phenomenon is due to the release of “free” iron from the iron complexes present in meat. As the concentration of “free” iron increased in red meat samples after pressure treatments [9,10], chelating agents (such as EDTA) effectively prevented the increased rates of oxidation seen in pressure-treated pork [6]. It is also possible that the effects of pressure may be related to changes in the integrity of the cell membrane [8].

### 3.2. Fatty Acids

#### 3.2.1. Fatty Acid Composition in Different Control Cuts

In this study, the fatty acid composition of saturated, monounsaturated, and polyunsaturated fatty acids in selected New Zealand lamb meat cuts is summarised in Table 1. The results in this study were similar to the fatty acid composition of other livestock species reared for meat production [30]. C16:0 palmitic acid and C18:0 stearic acid are usually the major saturated fatty acids in lamb meat and lamb meat products [35], and this was in agreement with our results for all cuts. Saturated fatty acids present in this study included C16:0, C17:0, C18:0, C20:0, C22:0 and C23:0 (Table 1). These fatty acids, except for C23:0, were significantly higher in the bolar cut, followed by the eye of loin, knuckle, tenderloin, and flat cuts. Similarly, Rhee et al. [31] who analysed the fatty acid composition of beef rump, tenderloin, inside and heel cuts, showed that C18:0 fatty acid in the tenderloin cut was significantly higher compared to rump, inside and heel cuts. Badiani et al. [32] analysed fresh and cooked beef bolar, flat and heel cuts, and reported that bolar cut contained the highest level of fatty acids compared to flat and heel cuts. In addition, the value of total saturated fatty acids in the bolar cut was about two times more than in flat and heel cuts in their study, which was similar to the results of this study.

The monounsaturated fatty acids in this project were C16:1, C17:1 and C18:1n-9. Kelly et al. [35] reported that oleic C18:1 was the most abundant monounsaturated fatty acid in meat products, similar to this study. The eye of loin cut had the highest level of C16:1, C17:1, C18:1n-9, and total monounsaturated fatty acids. Inside and rump cuts had a lower amount of C16:1, C17:1, and C18:1n-9 fatty acid and total monounsaturated fatty acid content compared to other cuts. Similarly, Badiani et al. [32] analysed fresh and cooked beef bolar, flat and heel cuts. They found that the monounsaturated fatty acids in the bolar cut were significantly higher than flat and heel cuts.

The polyunsaturated fatty acids reported in this study included 18:2n-6, 18:3n-6, 18:3n-3, 20:4n-6, 20:5n-3 and 22:2n-6 fatty acids. Kelly et al. [35] reported that linoleic C18:2 was the major unsaturated fatty acid in meat products, similar to this study for all cuts. Generally, tenderloin cut had the highest level of PUFAs. Lower but significant levels of PUFAs were present in flat, heel and inside cuts. Similarly, Manner et al. [36] reported significantly higher levels of C18:2 in the tenderloin cut than the heel cut in steers. Badiani et al. [32] also reported that total PUFAs in the beef bolar cut were significantly higher than flat and heel cuts. However, Rhee et al. [31] found no significant differences in PUFA levels in inside, heel, rump, and tenderloin cuts of porcine meat.

According to Wood et al. [30], the polyunsaturated/saturated fatty acid (PUFA/SFA) ratios for lamb are typically 0.1 but can be higher in some muscles. A higher PUFA to SFA ratio (≥0.4) is desirable as it decreases the risks of cardiovascular disease and metabolic syndrome [37]. Factors that affect this ratio include animal breed, sex and nutrition [38]. Our results showed that the PUFA/SFA ratios of heel (0.43), inside (0.53) and rump (0.51) cuts were higher than 0.4. The ratio of omega-6 to omega-3 PUFAs (n6: n3) shown in Table 1 is also important as it is a risk factor in cancer and coronary heart diseases [38]. The recommendation is for a ratio of less than 4 [30]. The n6:n3 ratios in this study ranged from 2.65 to 3.90 for all samples, which fall within the recommended ratio.

#### 3.2.2. Effect of Different HPP Treatments on Fatty Acid Composition

HPP-treated rump and heel cuts had significantly higher SFA and MUFA contents, as well as significantly lower PUFA/SFA ratios compared to control samples (Appendix A). As palmitic acid (C16:0) and stearic acid (C18:0) are the most abundant SFAs, the increase in SFAs is likely to be attributed to these fatty acids [39,40] while the significant increase in MUFAs is likely to be associated with a significant increase of C18:1n-9 since lamb meat has been reported to contain more oleic acid, C18:1n-9 [41,42]. Similarly, in terms of cuts, Ma et al. [40] reported that the shoulder cut of lamb had significantly higher MUFA content compared to the loin cut. This increase was also attributed to the increase in C18:1n-9 fatty acid. PUFA content was not significantly different to control for both cuts, suggesting that little or no oxidation occurred [43]. This is supported by the oxidation results in this study for rump and heel cuts, which reached a maximum of 0.448 and 0.541 mg MDA/kg, respectively, at 600 MPa.

SFA and MUFA content were significantly lower in HPP-treated inside, bolar, knuckle, eye of loin, tenderloin and flat cuts at all pressures compared to control samples. PUFA content was also lower in HPP-treated inside (all pressures), knuckle (400 and 600 MPa), eye of loin (300 MPa) and tenderloin (600 MPa) cuts compared to control. Similarly, Yang et al. [44] found that total SFA, MUFA and PUFA content significantly decreased during HPP treatment of marinated pork meat compared to control. He et al. [45] also found that the percentage of SFA and MUFA significantly decreased after HPP treatment (350 and 500 MPa) of pork and subsequent storage. Yang et al. [46] reported a significant decrease in PUFA in dry-cured ham in the first 4 months of aging. With respect to individual fatty acids, a significant decrease in PUFAs at high pressure was mainly associated with changes in C18:2n-6 and C18:3n-3. Similarly, Ma et al. [40] found that total PUFA content was significantly lower at 200 and 300 MPa in shank and shoulder cuts of lamb compared to control samples. The authors associated this decrease in PUFA content with changes in C18:2n-6 and C18:3n-3 fatty acid content. Oxidation of long-chain fatty acids is slower, with unsaturated fatty acids being oxidized more rapidly than saturated fatty acids [47]. In this study, as pressure increased, lipid oxidation levels also increased. Shahidi and Zhong [28] demonstrated that PUFAs in meat can react with molecular oxygen through a free radical chain mechanism to form fatty acyl hydroperoxides and other primary oxidation products. In addition, Wood et al. [30] indicated that oxidative stability PUFAs is affected by the composition of fatty acids during processing, ageing and retail display. Pereda et al. [48] further showed that fatty acids can decrease as a result of fatty acid oxidation and acidification, thereby supporting the decrease in fatty acid content in this study.

As mentioned before, the recommended PUFA/SFA ratio should be above 0.4 [30]. The PUFA/SFA ratios for different muscles ranged from 0.16 to 0.49 in goat meat, according to Banskalieva et al. [12]. Tshabalala et al. [49] further reported PUFA/ SFA ratios of 0.62 to 0.79 in goat meat. In this study, HPP-treated inside (at all pressures), knuckle (300 and 400 MPa), eye of loin (300 MPa), tenderloin (400 MPa) and flat (all pressures) muscles had PUFA/SFA ratios of more than 0.4 compared to control samples. Hence, HPP processing in this study resulted in a positive effect on PUFA/SFA ratios of some lamb cuts. Furthermore, the n:6/n:3 PUFA ratios of all samples in this study remained within the recommended level of ≤4 [38]. Similarly, McArdle et al. [18] reported that high pressure had no significant effect on n6:n3 ratios in lamb meat.

### 3.3. Free Amino Acids

#### 3.3.1. Free Amino Acid Content of Different Control Cuts

In this study, seventeen amino acids (alanine, glycine, valine, leucine, threonine, serine, isoleucine, proline, aspartic acid, methionine, glutamic acid, phenylalanine, ornithine, lysine, histidine, tyrosine, tryptophan) were identified and quantified. The results are given as mean values (mg/100 g) of free amino acids, as shown in Table 2. The presence of cysteine (CYS) and arginine (ARG) have been previously reported in raw goat meat at 0.01 and 12.2 mg/100 g, respectively [50], which are quite low values. However, these free amino acids were not identified in the current study due to the limitations of the amino acid analysis kit used (Phenomenex, 2003), as reported in another study [51].

Nine essential free amino acids (valine, leucine, isoleucine, methionine, phenylalanine, lysine, histidine, tyrosine and tryptophan), and eight non-essential free amino acids (alanine, glycine, threonine, serine, proline, aspartic acid, glutamic acids and ornithine) were detected. In general, the type of cut had a significant effect on the free amino acid composition. Table 2 showed that the total amount of free amino acids in the inside cut was significantly higher for almost all individual free amino acids except for proline, glutamic acid and methionine. Bolar, eye of loin, heel, knuckle, rump and tenderloin cuts, on the other hand, had significantly lower total free amino acids. Generally, our results showed that alanine, glycine, glutamic and aspartic acid are the major non-essential amino acids. Valine and leucine were the major essential amino acids in this study. Similarly, Holló et al. [52] reported that the highest essential amino acid fractions in beef were lysine and leucine.

#### 3.3.2. Non-essential Free Amino Acid Content in Different Control Cuts

The major non-essential free amino acids found were alanine, glycine, glutamic acid and aspartic acid (Table 2). Madruga et al. [50] reported that the most abundant free amino acids in the rump of goat meat were glycine, alanine, and glutamine. Watanabe et al. [53] reported that alanine, glutamic acid and aspartic acid were the major amino acids in cattle. In addition, Holló et al. [52] reported that the major non-essential amino acids in beef were glutamic acid and aspartic acid. The eight non-essential free amino acids (alanine, glycine, threonine, serine, proline, aspartic acid, glutamic acids and ornithine) found in this study accounted for approximately 70% to 80% of the total amino acids.

In inside and flat cuts, the total non-essential free amino acids values were significantly higher compared to other cuts. On the other hand, eye of loin, rump and tenderloin cuts had significantly lower total non-essential free amino acid content. In this study, the flat cut had a significantly higher level of glutamic acid, and the rump cut had a significantly lower level of glutamic acid content. Similarly, Aristoy and Toldrá [54] analysed the free amino acids in porcine skeletal muscle with different oxidative patterns and demonstrated that the content of non-essential free amino acids such as glutamic acid and proline, as well as total free amino acids in the trapezius muscles (flat), were significantly higher than in rump muscles.

Franco et al. [55] studied total amino acids and free amino acids in different beef muscles (heel, bolar, inside, eye of loin, masseter and cardiac muscles), and reported that glutamic acid was the highest in the heel cut compared to other cuts (bolar, inside, eye of loin), similar to this study. In fact, our result showed that the knuckle, flat and heel cuts had significantly higher glutamic acid content than other muscles (eye of loin, inside, bolar, tenderloin and rump). Franco et al. [55] also reported that aspartic acid was the major non-essential amino acid in all muscles (heel, bolar, inside, and eye of loin cuts). Similarly, aspartic acid was the major non-essential amino acid in this study, with significantly higher amounts in bolar, eye of loin, flat and inside cuts. Meanwhile, rump and tenderloin cuts had the lowest amount of aspartic acid in this study.

#### 3.3.3. Essential Free Amino Acids Content in Different Control Cuts

In this study, nine essential free amino acids (valine, leucine, isoleucine, methionine, phenylalanine, lysine, histidine, tyrosine and tryptophan) accounted for 20% to 30% of total amino acids. The major essential amino acids found were valine, leucine, phenylalanine, tyrosine and histidine. Madruga et al. [50] reported that the most abundant essential free amino acids in the rump of goat meat were leucine and valine. As seen in Table 2, inside cut had significantly higher essential amino acids (EAAs), followed by the rump cut. Bolar, eye of loin, and heel cuts had the least EAA content.

There were significant differences in leucine, isoleucine and methionine content between cuts. Leucine content was highest in the inside cut (27.67 mg/100 g). However, bolar (12.07 mg/100 g), eye of loin (10.71 mg/100 g), heel (12.68 mg/100 g) and tenderloin (14.06 mg/100 g) cuts had significantly lower leucine content, similar to porcine rump cut (19.15 mg/100 g), as reported by Cornet and Bousset [56]. However, results in this study showed a higher leucine concentration for all cuts compared to Madruga et al. [50] (7.9 mg/100 g in raw goat rump cut) and Franco et al. [55] (beef heel (6.00 mg/100 g), bolar (2.90 mg/100 g), inside (2.40 mg/100 g), and eye of loin (1.55 mg/100 g)).

Isoleucine content was highest in inside (23.36 mg/100 g) and rump (23.00 mg/100 g) cuts. Similarly, Cornet and Bousset [56] reported 26.98 mg/100 g isoleucine in porcine rump cut. Bolar (7.8 mg/100 g) and eye of loin (10.58 mg/100 g) cuts had the lowest level of isoleucine content. These results were higher than reported by Franco et al. [55] for beef heel (4.8 mg/100 g), bolar (5.02 mg/100 g), inside (5.55 mg/100 g), and eye of loin (2.61 mg/100 g) cuts.

Methionine was only present at a very low concentration in this study. It was highest in the rump cut (14.78 mg/100 g). Bolar (4.62 mg/100 g), eye of loin (4.95 mg/100 g), flat (4.11 mg/100 g), heel (3.53 mg/100 g) and tenderloin (4.74 mg/100 g) cuts were present at significantly lower levels. These findings are similar to the methionine content reported by Cornet and Bousset [56], who reported 10.06 mg/100 g methionine in porcine rump cut, and Franco et al. [55], who reported 4.0 mg/100 g in beef heel cut, 1.55 mg/100 g in beef bolar cut, 1.77 mg/100 g in inside cut, and 0.87mg /100 g in eye of loin cut. In other red meats, such as camel [57], hen [57] and ostrich [58], lysine was the major essential free amino acid. In our results, lysine was not significantly different between all cuts.

#### 3.3.4. Effect of Different HPP Treatments on Free Amino Acid Content

The effect of high-pressure treatment on the free amino acid content is shown in Appendix A. Suzuki et al. [2] suggested that free amino acids have an important role in determining brothy and meaty flavours and that they are precursors of meat flavour. However, the effect of HPP on free amino acid content is not known. Campus et al. [22] studied free amino acids in dry-cured loins, and they showed that high pressure (300 to 400 MPa for 10 min at 20 °C) can stabilize the free amino acid content during storage due to a reduction in the activity of amino peptidases. Moreover, Suzuki et al. [2] reported that high-pressure treatments (200 to 400 MPa at ambient temperature) did not influence the amount of amino acids in beef shoulder skeletal muscles (heel).

As seen in Appendix A, HPP significantly increased the total free amino acid composition, compared to control, at all pressures for almost all cuts except the inside and eye of loin cuts. Many researchers have suggested that processing meat would increase the presence of certain free amino acids by proteolysis [54]. Most cuts had the highest levels of total amino acids at the 600 MPa treatment except for the inside, eye of loin and tenderloin cuts. Total amino acids were significantly higher in the tenderloin cut at 200 and 400 MPa and the inside cut at 200 MPa compared to control. The increase in free amino acids was similarly reported by Ohmori et al. [21], who suggested that high pressure between 100 and 300 MPa for 10 min at 25 °C increases the overall autolytic activity of raw beef round (inside cut) meat and leads to a higher concentration of free amino acids. However, their results also showed that with higher pressure treatments at 400 and 500 MPa, the concentration of free amino acids was identical to that of the control sample, unlike results from this study.

As for the eye of loin cut, a significant decrease in total amino acids was observed at 600 MPa. In fact, there was a significant decrease in the essential amino acids leucine and isoleucine. Simonin et al. [20] suggested that muscle proteins are vulnerable to oxidative reactions that result in the loss of EAAs and a decrease in protein digestibility. The mechanisms and reaction pathways for lipid and protein oxidations are different, but they are directly linked; both processes may be affected by similar prooxidant and antioxidant factors. Indeed, it seems that protein oxidation is observed under the same pressure levels as those found when lipid oxidation occurs (>300 MPa). Results of lipid oxidation in this study support this and, in fact, the eye of loin cut had the highest level of lipid oxidation at 0.905 mg MDA/kg. In conclusion, our results showed that HPP processing of all cuts, except for eye of loin, generally increased the concentration of free amino acids. Further studies on the digestibility of meat [59] and HPP process optimisation to improve meat quality can contribute to the application of HPP in the real-life industrial processing of meat. Our study also promises the wider application of non-thermal processing technology. For example, the use of Pulsed Electric Field (PEF) in the seafood category [60]. 

### 3.4. Multiple Factor Analysis on Fatty Acids, Free Amino Acid Content and TBARS

Figure 2 illustrates the MFA plot showing the interrelationship between fatty acid, free amino acid and TBARS measures with varying HPP pressure treatments of different lamb meat cuts. Fatty acid and free amino acid scores showed an inverse relationship in this study. Most of the sample cuts were separated along the F1 axis, with 33.14% of variance explained. For most cuts (flat, heel, inside, knuckle, and rump), increasing HPP pressure treatments were associated with an increase in most free amino acids that had high positive loadings along the F1 axis. Previous literature has reported that dry-cured meat products can generate a large amount of small peptides and free amino acids, which can directly develop taste characteristics [61], aroma attributes and flavour precursors [62]. Koutsidis et al. [62] reported that free amino acids such as leucine, isoleucine, serine, threonine, valine, and phenylalanine may contribute to the brown/roasted attribute, which is desired by consumers. In a study by Herranz et al. [63], dry-fermented sausages treated with a 0.159% mixture of valine, leucine and isoleucine (58/35/66; *w*/*w*) received the highest score for odour, flavour contribution and overall quality. Better flavour scores of the treated samples can be attributed to the higher concentration of volatile compounds that are derived from free amino acids, especially Ile, Leu, and Val. Similarly, Ma et al. [40] reported that lamb meat cuts treated with HPP at 400 MPa had significantly higher His, Leu, Met, Lys, and Pro content, and samples treated with HPP at 600 MPa were higher in Phe, Tyr, Gly, and Ile content compared to control. In another study by Yang et al. [64], HPP-treated (150 MPa) samples of marinated meat in soy sauce had a significantly higher concentration of glutamate and alanine than control, which contributed to the umami and sweet taste, respectively, that may further enhance the flavour of the meat. Although free amino acid content results in an increase in sensory quality, high levels of free amino acids, on the other hand, may result in decreased sensory acceptability of the product due to excessive proteolysis. In a study by Herranz et al. [63] samples treated with a 1.01% pool of 20 free amino acids (gly/asn/his/arg/thr/ala/pro/tyr/val/met/ile/leu/phe/trp/lys/asp/glu/ser/gln/cys) received lower scores for flavour compared to samples treated with 0.159% of amino acids (val/leu/ile), indicating that increased concentration of free amino acids may decrease the sensory quality of a product. The changes in chemical composition with HPP processing necessitate further work that investigates the sensory quality of the high-pressure-processed samples utilizing sensory temporal methods such as Temporal Dominance of Sensations [65] or Check-All-That Apply [66].

Control and low-pressure-treated bolar, eye of loin, and tenderloin lamb cuts were associated with an increase in most fatty acids that had high negative loadings along the F1 axis. Increased pressure treatments of these cuts were associated with a decrease in fatty acids that had low negative loadings along the F1 axis. Bolar, eye of loin and tenderloin cuts were significantly lower in SFA, MUFA, and PUFA content at higher pressure treatments, as compared to control (Table 1). A significant decrease in the fatty acids in the loin cut as compared to shank and shoulder cuts of lamb at higher pressures has also been observed by Ma et al. [40]. This reduction can be attributed to the increased oxidation and acidification of the fatty acids at higher pressures.

The tenderloin and eye of loin lamb cuts treated at high pressures of 400 and 600 MPa were associated with higher TBARS values and separated from samples treated at 200 and 300 MPa along the F2 axis, which accounted for 22.18% of the variance. Higher pressure treatments resulted in higher TBARS values in tenderloin and eye of loin lamb cuts, indicating their decreased oxidative stability. Park et al. [29] reported that pork loins had higher FFA (free fatty acid) values than belly meats, suggesting their increased oxidative susceptibility. Interestingly, tenderloin had a high PUFA content (Table 1), which makes it prone to lipolysis. Lipolysis has been regarded as a promoter of lipid oxidation due to the aggregation of fatty acids, particularly PUFAs, which are prone to lipid peroxidation [29]. All these findings resonate with the results described in Section 3.2.1.

Results from this study highlight that varying HPP treatments influenced the chemical composition of different lamb cuts. As these changes in chemical composition may affect the sensory qualities of various lamb cuts, further research is required to understand how HPP treatment can influence the sensory and flavour composition of HPP-treated lamb meat.

## 4. Conclusions

In this study, HPP treatments, varying in pressure, were applied to eight different lamb cuts in order to understand how important flavour indicators like lipid oxidation, fatty acid and free amino acid content were influenced. The results indicated that, in general, the eye of loin and tenderloin cuts treated at 400 and 600 MPa and the flat and heel cuts treated at 600 MPa are not suitable for the application of HPP. For most cuts, increasing the pressure of HPP treatments was associated with an increase in most of the free amino acids, a decrease in fatty acids, and higher TBARS values. The results from this study will be useful to the meat industry, with the increasing interest in the area of HPP to process meat products for export to the global market. It is important to note that in this study, a series of pilot tests of samples was used to determine the sampling method, using classical statistical inference (e.g., ANOVA). Hence, the limitation inherent to this approach is the lack of power calculation and understanding of effect sizes. We recommend future studies to utilize design of experiment approach to overcome the aforementioned statistical limitations [67].

## Figures and Tables

**Figure 1 foods-09-01444-f001:**
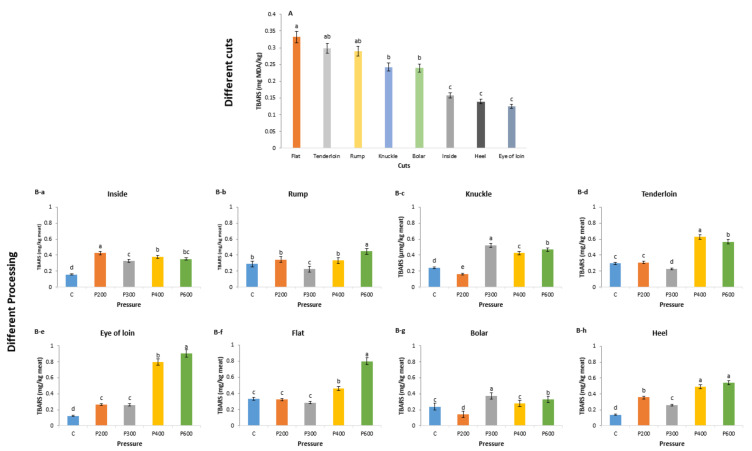
Lipid oxidation marker (thiobarbituric acid reactive substances (TBARS) in different lamb cuts and high-pressure processing samples. (**A**) different cuts and (**B**) different processing means of TBARS value with different cut samples (**B-a**–**B-h**) differ significantly using Fisher’s least significant difference (*p* < 0.05).

**Figure 2 foods-09-01444-f002:**
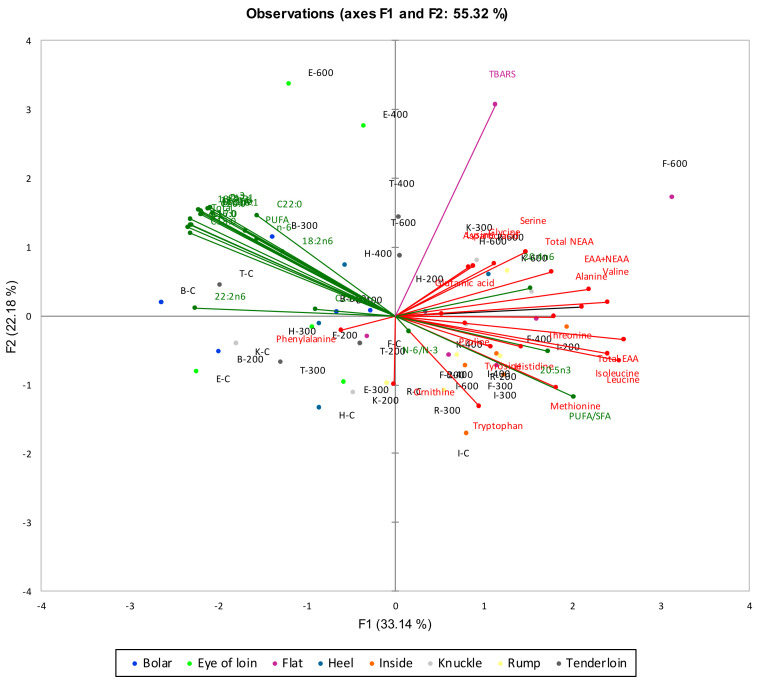
Multiple factor analysis (MFA) biplot for amino acids (red) and free fatty acids (green) for samples varying in cuts and HPP treatments. F1 (33.14%) and F2 (22.18%) explained a total of 55.32% of the variance.

**Table 1 foods-09-01444-t001:** Fatty acid composition (mg/100g) in different New Zealand lamb meat cuts.

Fatty Acids/Cuts	Bolar	Eye of Loin	Flat	Heel	Inside	Knuckle	Rump	Tenderloin	*p*-Value
C16:0	14.96 ± 0.03 ^a^	14.08 ± 0.09 ^b^	8.02 ± 0.61 ^d^	5.14 ± 0.11 ^e^	3.8 ± 0.16 ^f^	10.53 ± 0.02 ^c^	4.5 ± 0.44 ^e^	10.84 ± 0.29 ^c^	****
C16:1	0.77 ± 0.03 ^b^	0.9 ± 0.01 ^a^	0.56 ± 0.05 ^e^	0.39 ± 0.02 ^f^	0.31 ± 0.01 ^g^	0.62 ± 0.01 ^d^	0.35 ± 0.02 ^fg^	0.71 ± 0.00 ^c^	****
C17:0	1.06 ± 0.00 ^a^	0.82 ± 0.01 ^b^	0.44 ± 0.04 ^e^	0.35 ± 0.01 ^f^	0.27 ± 0.00 ^g^	0.67 ± 0.01 ^d^	0.32 ± 0.00 ^f^	0.73 ± 0.05 ^c^	****
C17:1	0.48 ± 0.00 ^a^	0.46 ± 0.01 ^a^	0.31 ± 0.01 ^cd^	0.25 ± 0.00 ^de^	0.21 ± 0.01 ^e^	0.37 ± 0.01 ^bc^	0.22 ± 0.01 ^e^	0.41 ± 0.09 ^ab^	****
C18:0	15.86 ± 0.21 ^a^	11.33 ± 0.06 ^bc^	5.58 ± 0.59 ^d^	4.63 ± 0.15 ^e^	3.2 ± 0.23 ^f^	10.86 ± 0.05 ^c^	4.58 ± 0.17 ^e^	11.63 ± 0.18 ^b^	****
18:1n9	16.29 ± 0.03 ^b^	20.28 ± 0.12 ^a^	12.77 ± 0.69 ^d^	8.27 ± 0.3 ^e^	6.13 ± 0.26 ^f^	14.53 ± 0.05 ^c^	6.89 ± 0.72 ^f^	16.29 ± 0.5 ^b^	****
18:2n6	2.6 ± 0.09 ^de^	3.48 ± 0.22 ^c^	2.28 ± 0.27 ^e^	2.78 ± 0.1 ^d^	2.43 ± 0.22 ^de^	3.96 ± 0.03 ^b^	3.19 ± 0.07 ^c^	5.64 ± 0.15 ^a^	****
18:3n6	0.68 ± 0.03 ^a^	0.49 ± 0.06 ^b^	0.21 ± 0.05 ^c^	0.23 ± 0.00 ^c^	0.15 ± 0.04 ^c^	0.39 ± 0.02 ^b^	0.24 ± 0.05 ^c^	0.4 ± 0.08 ^b^	****
18:3n3	1.29 ± 0.02 ^b^	1.1 ± 0.02 ^c^	0.72 ± 0.03 ^de^	0.71 ± 0.00 ^e^	0.64 ± 0.03 ^f^	1.12 ± 0.01 ^c^	0.76 ± 0.03 ^d^	1.38 ± 0.02 ^a^	****
C20:0	0.53 ± 0.02 ^a^	0.41 ± 0.02 ^b^	0.27 ± 0.03 ^c^	0.26 ± 0.03 ^c^	0.24 ± 0.01 ^c^	0.39 ± 0.01 ^b^	0.28 ± 0.02 ^c^	0.41 ± 0.01 ^b^	****
20:4n6	0.25 ± 0.00	0.28 ± 0.09	0.35 ± 0.12	0.37 ± 0.09	0.41 ± 0.01	0.41 ± 0.02	0.45 ± 0.01	0.44 ± 0.05	ns
20:5n3	0.18 ± 0.00 ^d^	0.27 ± 0.00 ^c^	0.32 ± 0.03 ^ab^	0.33 ± 0.00 ^ab^	0.33 ± 0.03 ^ab^	0.31 ± 0.01 ^b^	0.34 ± 0.00 ^ab^	0.34 ± 0.01 ^a^	****
C22:0	0.24 ± 0.01	0.2 ± 0.01	0.19 ± 0.02	0.18 ± 0.00	0.17 ± 0.00	0.22 ± 0.01	0.19 ± 0.00	0.23 ± 0.05	ns
22:2n6	0.39 ± 0.01 ^a^	0.32 ± 0.01 ^b^	0.21 ± 0.05 ^cde^	0.2 ± 0.03 ^de^	0.14 ± 0.01 ^f^	0.26 ± 0.02 ^c^	0.19 ± 0.02 ^ef^	0.25 ± 0.01 ^cd^	****
C23:0	0.2 ± 0.00	0.1 ± 0.18	0.12 ± 0.12	0.12 ± 0.02	0.14 ± 0.00	0.24 ± 0.03	0.23 ± 0.00	0.1 ± 0.08	ns
SFA	32.85 ± 0.21 ^a^	26.92 ± 0.33 ^b^	14.62 ± 1.4 ^d^	10.69 ± 0.31 ^e^	7.82 ± 0.40 ^f^	22.91 ± 0.07 ^c^	10.09 ± 0.63 ^e^	23.94 ± 0.55 ^c^	****
MUFA	17.53 ± 0.00 ^b^	21.65 ± 0.10 ^a^	13.64 ± 0.75 ^d^	8.91 ± 0.32 ^e^	6.65 ± 0.28 ^f^	15.53 ± 0.05 ^c^	7.45 ± 0.75 ^f^	17.4 ± 0.59 ^b^	****
PUFA	5.4 ± 0.15 ^cd^	5.93 ± 0.35 ^bc^	4.09 ± 0.54 ^f^	4.62 ± 0.22 ^ef^	4.1 ± 0.26 ^f^	6.46 ± 0.05 ^b^	5.16 ± 0.07 ^de^	8.45 ± 0.32 ^a^	****
n-3	1.48 ± 0.02 ^b^	1.36 ± 0.02 ^c^	1.04 ± 0.05 ^d^	1.04 ± 0.00 ^d^	0.97 ± 0.01 ^e^	1.43 ± 0.00 ^b^	1.1 ± 0.03 ^d^	1.73 ± 0.03 ^a^	****
n-6	3.92 ± 0.13 ^d^	4.57 ± 0.37 ^bc^	3.04 ± 0.49 ^e^	3.58 ± 0.22 ^de^	3.13 ± 0.26 ^e^	5.03 ± 0.05 ^b^	4.06 ± 0.10 ^cd^	6.73 ± 0.29 ^a^	****
PUFA/SFA	0.16 ± 0.00 ^f^	0.22 ± 0.02 ^e^	0.28 ± 0.01 ^d^	0.43 ± 0.01 ^b^	0.53 ± 0.01 ^a^	0.28 ± 0.00 ^d^	0.51 ± 0.01 ^a^	0.35 ± 0.01 ^c^	****
n6:n3	2.65 ± 0.02 ^d^	3.35 ± 0.41 ^bc^	2.92 ± 0.21 ^cd^	3.43 ± 0.22 ^ab^	3.22 ± 0.23 ^bc^	3.5 ± 0.05 ^ab^	3.7 ± 0.15 ^ab^	3.9 ± 0.14 ^a^	***
total	55.77 ± 0.36 ^a^	54.5 ± 0.78 ^a^	32.34 ± 2.7 ^d^	24.21 ± 0.85 ^e^	18.57 ± 0.94 ^f^	44.9 ± 0.03 ^c^	22.71 ± 1.31 ^e^	49.79 ± 1.46 ^b^	****

Values with different superscripts (^a,b,c,d,e,f^) in the same row differ significantly within cuts. SFA stands for saturated fatty acids; MUFA stands for monounsaturated fatty acids; PUFA stands for polyunsaturated fatty acids. *p* < 0.0001 is presented as ∗∗∗∗ for level of significance; *p* < 0.001 is presented as ∗∗∗ for level of significance; ns means not statistically significant.

**Table 2 foods-09-01444-t002:** Free amino acid composition (mg/100g) in different New Zealand lamb meat cuts.

Cuts	Bolar	Eye of Loin	Flat	Heel	Inside	Knuckle	Rump	Tenderloin	*p*-Value
**Nonessential**									
Alanine	62.79 ± 0.67^c^	54.85 ± 2.14^cd^	54.3 ± 2.36^d^	78.25 ± 8.27^b^	92.89 ± 2.81^a^	58.54 ± 0.11^cd^	78.65 ± 2.48^b^	72.75 ± 3.61^b^	****
Glycine	60.45 ± 2.37^a^	44.74 ± 3.95^c^	49.97 ± 2.63^bc^	49.7 ± 1.32^bc^	59.44 ± 4.61^a^	44.58 ± 1.37^c^	54.69 ± 4.37^ab^	43.82 ± 1.59^c^	**
Serine	12.91 ± 2.67^c^	17.12 ± 4.37^bc^	26.01 ± 1.39^a^	17.92 ± 4.96^bc^	26.26 ± 4.74^a^	12.41 ± 2.16^c^	18.28 ± 2.8^bc^	21.2 ± 0.22^ab^	*
Threonine	11.93 ± 1.49^c^	15.54 ± 6.92^bc^	27.05 ± 0.75^a^	15.31 ± 2.49^bc^	28.76 ± 1.06^a^	15.4 ± 0.85b^c^	18.98 ± 0.14^b^	17.6 ± 0.16^bc^	**
Proline	9.94 ± 1.42	8.08 ± 1.65	12.12 ± 2.57	5.84 ± 1.1	7.44 ± 0.34	6.12 ± 1.52b	8.7 ± 3.24	3.1 ± 0.85	ns
Glutamic acid	63.57 ± 6.65^bc^	54.66 ± 0.13^c^	82.66 ± 4.75^a^	78.02 ± 18.46^ab^	56.76 ± 3.68^c^	82.16 ± 6.19^a^	35.35 ± 2.89^d^	48.44 ± 1.71^cd^	**
Aspartic acid	122.33 ± 1.89^ab^	114.73 ± 1.5^abc^	129.87 ± 2.86^a^	100.96 ± 19.5^cd^	118.98 ± 1.37^abc^	105.54 ± 3.07^bcd^	71.06 ± 9.71^e^	89.32 ± 1.37^de^	***
Ornithine	4.93 ± 0.06	5.94 ± 1.29	6.02 ± 0.76	4.94 ± 0.05	5.33 ± 0.97	5.45 ± 0.1	7.35 ± 2.19	4.51 ± 0.07	ns
NEAA	348.85 ± 6.75^bc^	315.66 ± 10.63^cde^	387.99 ± 4.6^a^	350.96 ± 38.51^b^	395.87 ± 2.88^a^	330.21 ± 9.24^bcd^	293.05 ± 3.16^e^	300.73 ± 0.24^de^	***
**Essential**									
Valine	13.85 ± 0.47^c^	19.61 ± 8.64^bc^	26.05 ± 4.4^b^	19.77 ± 0.1^bc^	38.01 ± 1.34^a^	18.67 ± 4.22^bc^	19.12 ± 1.21^bc^	16.88 ± 1.13^c^	**
Leucine	12.07 ± 0.58^cd^	10.71 ± 0.97^d^	15.29 ± 0.67^bc^	13.68 ± 2.05^bcd^	27.67 ± 1.84^a^	15.2 ± 3.33^bc^	16.71 ± 0.95^b^	14.06 ± 0.5^bcd^	****
Isoleucine	7.8 ± 0.28^e^	10.58 ± 2.46^de^	16.87 ± 0.66^b^	13.15 ± 1.95^bcd^	23.36 ± 0.5^a^	15.44 ± 1.87^bc^	23 ± 2.65^a^	12 ± 0.69^cd^	****
Methionine	4.67 ± 0.32^d^	4.95 ± 1.01^d^	4.11 ± 0.94^d^	3.53 ± 0.65^d^	9.38 ± 0.24^b^	6.96 ± 1.33^c^	14.78 ± 0.48^a^	4.74 ± 0.42^d^	****
Phenylalanine	13.05 ± 1.49^cde^	10.13 ± 1.5^e^	11.55 ± 0.23^de^	16.61 ± 1.94^ab^	19.91 ± 0.75^a^	13.1 ± 2.11^cde^	14.8 ± 1.09^bcd^	15.4 ± 1.54^bc^	**
Lysine	8.26 ± 0.26	10.25 ± 1.61	9.62 ± 0.66	8.67 ± 0.97	9.51 ± 0.41	8.24 ± 0.59	9.29 ± 0.33	10.97 ± 0.12	ns
Histidine	13.67 ± 0.75	10.64 ± 0.25	13.24 ± 0.42	11.81 ± 1.27	12.57 ± 1.38	12.66 ± 1.13	11.87 ± 0.23	12.9 ± 1.35	ns
Tyrosine	10.43 ± 1.44^b^	16.98 ± 2.4^a^	11.23 ± 0.5^b^	9.14 ± 1.08^b^	17.12 ± 0.71^a^	11.27 ± 2.58^b^	11.5 ± 2.62^b^	15.54 ± 0.78^a^	**
Tryptophan	5.68 ± 0.51^d^	8.03 ± 0.51^ab^	6.24 ± 0.6^cd^	5.17 ± 0.77^d^	8.63 ± 0.02^a^	7.51 ± 0.4^ab^	7.16 ± 0.07^bc^	5.23 ± 0.54^d^	***
EAA	89.48 ± 1.19^d^	101.88 ± 9.49^cd^	114.19 ± 8.09^bc^	101.52 ± 6.87^cd^	166.17 ± 1.41^a^	109.04 ± 16.37^bc^	128.22 ± 6.62^b^	107.71 ± 7.08^cd^	****
TOTAL	438.33 ± 7.95^c^	417.54 ± 20.12^c^	502.18 ± 3.49^b^	452.48 ± 45.39^c^	562.03 ± 1.46^a^	439.25 ± 25.61^c^	421.28 ± 9.78^c^	408.44 ± 7.32^c^	***

Values with different superscripts (^a,b,c,d,e^) in the same row differ significantly within cuts. EAA stands for essential free amino acids; NEAA stands for nonessential free amino acids. *p* < 0.0001 is presented as ∗∗∗∗ for 0.001% level of significance; *p* < 0.001 is presented as ∗∗∗ for 0.01% level of significance; *p* < 0.01 is presented as ∗∗ for 1% level of significance; *p* < 0.05 is presented as ∗ for 5% level of significance; ns means not statistically significant.

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
