# Peer review of "Effect of High Hydrostatic Pressure Processing on the Chemical Characteristics of Different Lamb Cuts"

_foods, 2020, doi:10.3390/foods9101444_

Round 1

Reviewer 1 Report

The study titled “Effect of High Hydrostatic Pressure Processing on the Chemical Characteristics of Different Lamb Cuts” is discussing about the processing effect on lamb cuts. I would give some suggestions to authors to improve this article.

Abstract

The abbreviation, HPP, should be clearly defined and declared for only once. (Non-thermal high pressure processing? high pressure processing? high hydrostatic pressure processing?)

Introduction

In L41. Is the reference [4] precisely cited ? Please check.

In L45. The authors said that only few studies investigated the effects of HPP on fatty acid composition, however, you cited several articles thereafter. That is contradictory. And, your study seems not show the appropriate data related to fatty acid.

Materials and methods

In 2.2. HPP processing, the statements were inadequate for replication. More detail should be supplied.

In 2.3. Lipid oxidation, the same problem to 2.2. HPP processing, the statements were inadequate for replication, and more detail should be supplied.

In 2.4. The GC-FID should be combined in the latter half of this section.

In 2.5. The GC-FID should be described anew in the latter half of this section.

In 2.7. If possible, please add the operation detail of MFA.

Results and Discussion

In L175. TBA value or TBARS value ?

In 3.2.1. The statement of L213-219 should be presented in INTRODUCTION.

In L220, the abbreviation of fatty acid should be presented in where it was first used.

In 3.2.2. Supplementary Table 1 was not found in this manuscript. I think Supplementary Table 1 should be combined with Table 1.

In 3.3.4. Supplementary Table 2 & 3 were not found in this manuscript. I think Supplementary Table 2 & 3 should be combined with Table 2.

Since this study is titled “Effect of High Hydrostatic Pressure Processing on the Chemical Characteristics of Different Lamb Cuts”, the sections 3.2.1., 3.3.1. and 3.3.2. seem not so necessary. Moreover, the sections 3.2.1., 3.3.1. and 3.3.2. show no novel evidence in scientific study.   

[End]

Author Response

Reviewer 1

The study titled “Effect of High Hydrostatic Pressure Processing on the Chemical Characteristics of Different Lamb Cuts” is discussing about the processing effect on lamb cuts. I would give some suggestions to authors to improve this article.

We’d like to thank the reviewer, we have made amendments to the manuscript and have highlighted in yellow where we make revisions.

Abstract

The abbreviation, HPP, should be clearly defined and declared for only once. (Non-thermal high pressure processing? high pressure processing? high hydrostatic pressure processing?)

This has been corrected, the authors have only defined and declared it once in the beginning of the Abstract.

Introduction

In L41. Is the reference [4] precisely cited ? Please check.

We have correctly cited the part of the sentence to indicate that beef lipid changes was lower at 200 MPa.

In L45. The authors said that only few studies investigated the effects of HPP on fatty acid composition, however, you cited several articles thereafter. That is contradictory. And, your study seems not show the appropriate data related to fatty acid.

We have phrased this correctly to

Recent studies have investigated the effects of HPP on fatty acid composition.” 

Our study did indeed investigate fatty acid composition which can be found on Section 3.2

Materials and methods

In 2.2. HPP processing, the statements were inadequate for replication. More detail should be supplied.

More details has been added in the Section.

In 2.3. Lipid oxidation, the same problem to 2.2. HPP processing, the statements were inadequate for replication, and more detail should be supplied.

More details has been added in the Section.

In 2.4. The GC-FID should be combined in the latter half of this section.

This has been addressed accordingly

In 2.5. The GC-FID should be described anew in the latter half of this section.

This has been addressed accordingly

In 2.7. If possible, please add the operation detail of MFA.

MFA is a common statistical tool to analyse multiple datasets simultaneously which enables the investigation of relationships between datasets. As such MFA utilises PCA principles in our study due to its quantitative nature. PCA was carried out for each dataset in this case, amino acids, fatty acids, and TBARS. The first eigenvalues of each PCAs is then used to weight the all the dataset in the next analysis. After each PCA was carried out, a weighted PCA on all the dataset is then carried out.

Results and Discussion

In L175. TBA value or TBARS value ?

TBARS, this has been corrected accordingly

In 3.2.1. The statement of L213-219 should be presented in INTRODUCTION.

This has been shifted to Introduction on the second paragraph

In L220, the abbreviation of fatty acid should be presented in where it was first used.

The major fatty acids measured in this study and its abbreviation is now stated correctly in Introduction

In 3.2.2. Supplementary Table 1 was not found in this manuscript. I think Supplementary Table 1 should be combined with Table 1.

In 3.3.4. Supplementary Table 2 & 3 were not found in this manuscript. I think Supplementary Table 2 & 3 should be combined with Table 2.

Supplementary Table are rather vast – we have attached it to the manuscript submission system now and will await reviewer’s comments whether to include it in the main manuscript or not.

Since this study is titled “Effect of High Hydrostatic Pressure Processing on the Chemical Characteristics of Different Lamb Cuts”, the sections 3.2.1., 3.3.1. and 3.3.2. seem not so necessary. Moreover, the sections 3.2.1., 3.3.1. and 3.3.2. show no novel evidence in scientific study.   

We agree that this may not bring novel evidence in the scientific literature – however, we have used the control samples as a baseline when we do employ HPP technique.

Reviewer 2 Report

The manuscript “Effect of High Hydrostatic Pressure Processing on the  Chemical Characteristics of Different Lamb Cuts” is generally very well written and contains data of some relevance for a general readers as well as of high relevance for specialists in the topic. Although the subject of the paper could be of interest for the readers of the journal, the paper needs some corrections.

In general I think that is a lot more of the literature on the effect of HPP on the fatty acid composition and oxidative stability. In my opinion, the literature review should be extended.

  • Page 8, line 264:

Unfortunately, I have not found table 1 Supplementary anywhere. Therefore, I cannot assess the influence of HPP on the composition of fatty acids. I suggest including a table with the composition of fatty acids after HPP treatments in the main text. Alternatively, the total PUFA / MUFA / SFA content before and after HPP treatments can be presented on a figure.

  • Page 8, lines from 265 to 266:

Can you explain this sentence: “the increase in MUFA was related to the significant increase of 18:1n9 and 18:2n6”?  C18:2 is polyunsaturated fatty acid.

  • Page 8, line 270:

Why the content of SFA and MUFA were lower after HPP treatments?

  • Page 8, lines from 273 to 274:

Can you explain this sentence: “The decrease in PUFA was due to the decrease of mainly C18:1 n9 and C18:3 n3”? C18:1 is monosaturated fatty acid.

  • Page 8, lines from 287 to 288:

Unfortunately, there is no statistical analysis showing the influence of HPP on the composition of fatty acids and PUFA / SFA ratios. Individual results are also missing. On what basis do you conclude that HPP processing in this study resulted in a positive effect on PUFA / SFA ratios?

  • Page 11, line 370:

Unfortunately, I have not found table 2 & 3 Supplementary anywhere.

Therefore, I cannot assess the effect of different HPP treatments on free amino acids content.

The text also needs some editorial corrections, for example:

  • Page 1, line 59 and 67; Page 2, line 99 and 100: “â—¦C” to improve
  • Page 2, line 114 and 113; Page 2, line 130: missing spaces.
  • Reference no 20 to be improved.

Author Response

Reviewer 2

The manuscript “Effect of High Hydrostatic Pressure Processing on the Chemical Characteristics of Different Lamb Cuts” is generally very well written and contains data of some relevance for a general reader as well as of high relevance for specialists in the topic. Although the subject of the paper could be of interest for the readers of the journal, the paper needs some corrections.

We’d like to thank the reviewer, we have made amendments to the manuscript and have highlighted in yellow where we make revisions.

In general, I think that is a lot more of the literature on the effect of HPP on the fatty acid composition and oxidative stability. In my opinion, the literature review should be extended.

We have added more details in the Introduction section

Page 8, line 264:

Unfortunately, I have not found table 1 Supplementary anywhere. Therefore, I cannot assess the influence of HPP on the composition of fatty acids. I suggest including a table with the composition of fatty acids after HPP treatments in the main text. Alternatively, the total PUFA / MUFA / SFA content before and after HPP treatments can be presented on a figure.

We have not included the Supplementary Tables required for the manuscript. The Supplementary Tables are rather vast – we have attached it to the manuscript submission system now and will await reviewer’s comments whether to include it in the main manuscript or not.

Page 8, lines from 265 to 266:

Can you explain this sentence: “the increase in MUFA was related to the significant increase of 18:1n9 and 18:2n6”?  C18:2 is polyunsaturated fatty acid.

We have added a discussion on line 299-305. The line now reads

“As palmitic acid (C16:0) and stearic acid (C18:0) are the most abundant SFAs, the increase in SFA is likely to be attributed to these fatty acids [40, 41] while the significant increase in MUFA is likely to be associated with significant increase of C18:1n-9 since lamb meat has been reported to contain more elaidic acid, C18:1n-9 [42, 43]. Similarly, in terms of cuts Ma et al. [41] reported that shoulder cut of lamb had significantly higher MUFA content as compared to loin cut. This increase was also attributed to the increase in C18:1n-9 fatty acid.”

Page 8, line 270:

Why the content of SFA and MUFA were lower after HPP treatments?

We have added a discussion on line 309-321. The line now reads

“SFA and MUFA content were significantly lower in HPP treated inside, bolar, knuckle, eye of loin, tenderloin and flat cuts at all pressures compared to control samples. PUFA content was also lower in HPP treated inside (all pressures), knuckle (400 and 600 MPa), eye of loin (300 MPa) and tenderloin (600 MPa) cuts, compared to control. Similarly, Yang et al. [45] found total SFA, MUFA and PUFA content significantly decreased during HPP treatment of marinated pork meat compared to control. He et al. [46] also found that the percentage of SFA and MUFA significantly decreased after HPP treatment (350 and 500 MPa) of pork and subsequent storage. Yang et al. [47] reported a significant decrease in PUFA in dry-cured ham in the first 4 months of aging. With respect to individual fatty acids, a significant decrease in PUFA at high pressure was mainly associated with changes in C18:2n-6 and C18:3n-3. Similarly Ma et al. [41] found that total PUFA content was significantly lower at 200 MPa and 300 MPa in shank and shoulder cuts of lamb compared to control samples. The authors associated this decrease in PUFA content with changes in C18:2n-6 and C18:3n-3 fatty acids content.”

Page 8, lines from 273 to 274:

Can you explain this sentence: “The decrease in PUFA was due to the decrease of mainly C18:1 n9 and C18:3 n3”? C18:1 is monosaturated fatty acid.

This is linked to the previous comment that we have provided, please see above.

Page 8, lines from 287 to 288:

Unfortunately, there is no statistical analysis showing the influence of HPP on the composition of fatty acids and PUFA / SFA ratios. Individual results are also missing. On what basis do you conclude that HPP processing in this study resulted in a positive effect on PUFA / SFA ratios?

This result can now be found on Supplementary Table 1. We apologise for the inconvenience.

Page 11, line 370: 

Unfortunately, I have not found table 2 & 3 Supplementary anywhere. Therefore, I cannot assess the effect of different HPP treatments on free amino acids content. 

The Supplementary Tables are rather vast – we have attached it to the manuscript submission system now and will await reviewer’s comments whether to include it in the main manuscript or not. 

The text also needs some editorial corrections, for example: 

Page 1, line 59 and 67; Page 2, line 99 and 100: “â—¦C” to improve 

Page 2, line 114 and 113; Page 2, line 130: missing spaces. 

Reference no 20 to be improved. 

The recommended changes have been amended. 

Reviewer 3 Report

I enjoyed reading this manuscript; the needs of special groups of high hydrostatic pressure processing and lamb industry. This manuscript presents some interesting information including lipid oxidation, fatty acid and free amino acid content by the effect of 200-300 Mpa data. High hydrostatic pressure processing increased the total free amino acids composition and saturated, monounsaturated fatty acid content than those of atmosphere control samples. The results of this work could be used by the lamb processing industry and would help the preparation of specially treated HPP products. Abstract and introduction is well written and give enough information as expected by reader to know. Results and discussion cited some references to explain and confirm their observation. However, the discussion could be improved if the mechanism was elucidated more clear for the reader not in this field.

I have several suggestions to make and they are about the mistyping on the text as the attached file.

1) Five references should check their journal name in capital and wrong words should be revised in reference section.

2)The authors say that a “…500 Mpa in line 102 at page 2 should revised to 600 Mpa. I wonder if the information of 1500 g rpm at lines 120 and 121 is correct and it should be check again.

3) The data in line 175 should include inside cut? And The data in line 227 should delete inside from the sentence, because these fatty acids of inside is less than those of rump and heel cuts. These mistakes should be easy to correct and I expect that the manuscript can be accepted after minor revision.

Author Response

Reviewer 3

I enjoyed reading this manuscript; the needs of special groups of high hydrostatic pressure processing and lamb industry. This manuscript presents some interesting information including lipid oxidation, fatty acid and free amino acid content by the effect of 200-300 Mpa data. High hydrostatic pressure processing increased the total free amino acids composition and saturated, monounsaturated fatty acid content than those of atmosphere control samples. The results of this work could be used by the lamb processing industry and would help the preparation of specially treated HPP products. Abstract and introduction is well written and give enough information as expected by reader to know. Results and discussion cited some references to explain and confirm their observation. However, the discussion could be improved if the mechanism was elucidated more clear for the reader not in this field.

We’d like to thank the reviewer, we have made amendments to the manuscript and have highlighted in yellow where we make revisions.

I have several suggestions to make and they are about the mistyping on the text as the attached file.

1) Five references should check their journal name in capital and wrong words should be revised in reference section.

This has been fixed accordingly

2)The authors say that a “…500 Mpa in line 102 at page 2 should revised to 600 Mpa. I wonder if the information of 1500 g rpm at lines 120 and 121 is correct and it should be check again.

We apologise that this has created confusion, it is supposed to be 600MPa as to 500MPa.

3) The data in line 175 should include inside cut? And The data in line 227 should delete inside from the sentence, because these fatty acids of inside is less than those of rump and heel cuts.

The sentence on line 175 was for TBARS measurement so therefore this is correct. The fatty acids sentence on line 227 has been corrected as inside is significant from rump and heel. This can be found on line 260-262, the line now reads:

“These fatty acids except for C23:0 were significantly higher in the bolar cut, followed by eye of loin, knuckle, tenderloin, and flat cuts.“

These mistakes should be easy to correct and I expect that the manuscript can be accepted after minor revision.

Round 2

Reviewer 1 Report

This article has been well revised. The supplementary tables are truly vast, and I agree that the tables can be attached as  supplementary materials.

Author Response

Reviewer 1

This article has been well revised. The supplementary tables are truly vast, and I agree that the tables can be attached as  supplementary materials.

We’d like to thank the reviewer. We have included this as supplementary material.

Reviewer 2 Report

Thank you for the corrections to the manuscript “Effect of High Hydrostatic Pressure Processing on the Chemical Characteristics of Different Lamb Cuts”. In my opinion paper still needs some minor corrections.

  • Due to the fact that the title of the article is “Effect of High Hydrostatic Pressure Processing on the Chemical Characteristics of Different Lamb Cuts” I think that the paper should contain the results of fatty acid composition and free amino acids content before and after HPP.
  • Abstract, line 33: “Additionally, a significant increase in saturated, monounsaturated fatty acid, and free amino acid content was observed in HPP samples compared to control” - not for all samples.
  • Page 8, line 303: Are you sure you mean elaidic acid, C18:1n-9? It's the trans isomer of oleic acid.

The text still needs some editorial corrections, for example:

  • Please standardize the degrees Celsius, for example: Page 1, line 75, 80; Page 2, line 88, Page 3, line 105, 106, 121, 122, 125, 136, 137; Pages 4, line 152, 159, 174; Page 6, line 233.
  • Reference no 29 to be improved: (?10 ?C).

Author Response

Reviewer 2

Thank you for the corrections to the manuscript “Effect of High Hydrostatic Pressure Processing on the Chemical Characteristics of Different Lamb Cuts”. In my opinion paper still needs some minor corrections.

We’d like to thank the reviewer so far for the reviews.

Due to the fact that the title of the article is “Effect of High Hydrostatic Pressure Processing on the Chemical Characteristics of Different Lamb Cuts” I think that the paper should contain the results of fatty acid composition and free amino acids content before and after HPP.

We agree with the reviewer but we have already included the results for fatty and amino acid compositions of the samples in the Supplementary Tables and have also discussed the results in the appropriate sections.

Abstract, line 33: “Additionally, a significant increase in saturated, monounsaturated fatty acid, and free amino acid content was observed in HPP samples compared to control” - not for all samples.

It is true that this doesn’t occur in all cuts. We have rephrased it to majority of the HPP treated samples to avoid confusion. The sentence now reads:

“Additionally, a significant increase in saturated, monounsaturated fatty acid, and free amino acid content was observed in majority of HPP treated samples compared to control.”

Page 8, line 303: Are you sure you mean elaidic acid, C18:1n-9? It's the trans isomer of oleic acid.

The reviewer is correct, we have checked the reference and it is supposed to be oleic acid. We have made the appropriate change.

The text still needs some editorial corrections, for example:

Please standardize the degrees Celsius, for example: Page 1, line 75, 80; Page 2, line 88, Page 3, line 105, 106, 121, 122, 125, 136, 137; Pages 4, line 152, 159, 174; Page 6, line 233.

Reference no 29 to be improved: (?10 ?C).

We have changed the oC to a unicode standard of °C. For referencing, it is automatically updated unfortunately by a referencing software - we will notify the changes the journal's typesetters after the manuscript to change this.